# An Approach to Combine the Power of Deep Reinforcement Learning with a Graph Neural Network for Routing Optimization

**Bo Chen [1], Di Zhu [1], Yuwei Wang [2] and Peng Zhang [1,*]**

[1] Institute of Information Technology, PLA Information Engineering University, Zhengzhou 450001, China; daniel_0720@163.com (B.C.); sunyusheng2019@163.com (D.Z.)
[2] Institute of Acoustics, Chinese Academy of Sciences, Beijing 100089, China; wangyw@dsp.ac.cn
[*] Correspondence: zp@ndsc.com.cn

**Abstract:** Routing optimization has long been a problem in the networking field. With the rapid development of user applications, network traffic is continuously increasing in dynamicity, making optimization of the routing problem NP-hard. Traditional routing algorithms cannot ensure both accuracy and efficiency. Deep reinforcement learning (DRL) has recently shown great potential in solving networking problems. However, existing DRL-based routing solutions cannot process the graph-like information in the network topology and do not generalize well when the topology changes. In this paper, we propose AutoGNN, which combines a GNN and DRL for the automatic generation of routing policies. In AutoGNN, the traffic distribution in the network topology is processed by a GNN, while a DRL framework is used to train the parameters of neural networks without human expertise. Our experimental results show that AutoGNN can improve the average end-to-end delay of the network by up to 19.7% as well as present more robustness against topology changes.

**Keywords:** deep reinforcement learning; graph neural networks; software-defined networking; routing optimization





## 1. Introduction

With the rapid development of new information technologies, such as virtual reality (VR), 4K+ video, online conferences, and cloud services, among others, the information system infrastructure has recently come under a great burden of traffic transmission. On the other hand, such increases in network traffic require not only bandwidth expansion in the network infrastructure, but also call for better service quality, considering the different needs of various applications. Previous methods dealing with the expansion of network traffic have mainly relied on expanding the network scale and increasing the physical capacity of network devices, e.g., by deploying switches with more forwarding speed and cables with more capacity. Such investment in physical devices is both costly and inefficient. Therefore, there is a large space for improvement of the utility of the network devices.

In traditional computer networks, the functions of the network are usually integrated into each single device. Under such conditions, forwarding in the network layer depends on distributed algorithms to generate routing policies. These algorithms concentrate on the reachability and scalability of the network, of which "best-efforts" is the main idea. The emergence of software-defined networking (SDN) [1] has recently brought changes to this condition. With the centralized control logic of the network, the network traffic can be steered from a global view of the network, which can provide more information for the routing algorithm. However, designing a good algorithm or policy to control the traffic throughout the whole network is not trivial. The traffic distribution in the network is complicated and dynamically changes, which poses great challenges for traditional algorithms.

The prosperity of some certain network scenarios, such as data center networks (DCNs), also calls for better traffic scheduling in the network. Compared to traditional networks, DCNs usually experience a denser traffic and have more specific traffic characteristics. Furthermore, as the devices are usually placed in a more centralized manner in DCNs, the monitoring and managing of such network structures is easier, which leaves more space for network traffic optimization schemes to improve the performance of the network. Still, under such conditions, the performance of existing routing or traffic engineering schemes remains far from satisfactory [2].

To achieve a better utility of the network infrastructure under various network scenarios, many mathematical models have been recently proposed [2–6]. Such models mainly focus on a certain network scenario, then formulate an optimization problem to improve one or more network performance indices. Specifically, the formulated problems are usually NP-hard [3]. To ensure an efficient solution under fluctuating network traffic, heuristic solutions following such optimization problems are usually obtained, which cannot ensure the performance of the routing or traffic engineering schemes. Furthermore, as such models mainly focus on one certain network scenario, where the traffic exhibits a certain or clear characteristic, when the network experiences a change or update, the designed model may not be as effective.

In recent years, the development of machine learning technologies has brought new possibilities for the solution of complicated problems. The use cases of machine learning in natural language processing and image classification have demonstrated the power of state-of-the-art machine learning technologies, such as convolutional neural networks (CNNs) and recurrent neural networks (RNNs) [7]. The development of hardware technologies such as GPU computing has also propelled the advancement of machine learning technologies, by accelerating the associated computational processes. Therefore, machine learning-aided routing and traffic engineering technologies have been proposed.

In brief, there are three main types of machine learning technologies which can be applied to routing or traffic engineering, namely supervised learning, unsupervised learning, and reinforcement learning. Supervised learning uses a labelled data set to train the target algorithm in order to obtain an accurate model for a target problem. These methods usually achieve good performance in terms of the accuracy of classification of the input data, but a labelled data set is difficult to acquire in practice in the scenario of routing or traffic engineering. Unsupervised learning attempts to classify the network traffic without a labelled data set. However, such methods usually cannot reach good accuracy. Furthermore, both supervised learning and unsupervised learning methods only achieve primitive classification of the network traffic, requiring the use of other algorithms to make a step forward for the generation of routing or traffic engineering policies. In contrast, reinforcement learning (RL) methods can train a smart agent to automatically generate routing policies without labelled data, through a direct input of the network traffic. Furthermore, through use of the power of deep neural networks, an enhanced version named deep reinforcement learning (DRL) can reach the accuracy of supervised learning through the training framework of RL. Therefore, most state-of-the-art routing or traffic engineering schemes have adopted DRL as the main algorithm [8–11].

Despite this, there is still a problem in current DRL-based methods. The types of neural networks used in current DRL methods are mostly CNNs or RNNs [7]. Such neural networks are good at processing Euclidean-space data, such as images and text. However, network topology information is represented as graph-like data, which do not present Euclidean features. Therefore, the currently used neural networks are inappropriate for the processing of network topology information. Furthermore, when the network topology changes, CNNs and RNNs cannot generalize well to a new input size. Thus, such methods are vulnerable to topology changes.

To solve the problems related to the application of machine learning technologies in networks, in this paper, we combine the DRL and graph neural network (GNN) frameworks to generate routing policies for the communication networks. First, to better process the

graph-like topology information in the networks, we use a GNN to carry out the inference of traffic distribution information in the network. Then, to complete the training of the GNN, we employ a DRL framework to automatically tune the neural network parameters through the interaction of the algorithm and the network. The output of the GNN network is a list of values, each of which corresponds to a link weight. With this link weight, the routing path of each flow can be calculated through a weighted shortest path algorithm. The contributions of this paper can be summarized as follows:

1.   We propose an automatic routing mechanism that uses DRL to dynamically control the forwarding paths of the flows in the network.
2.   We design a GNN-based neural network system to process traffic information directly based on the graph-like network topology data, which can also generalize well to topology changes in the network.
3.   We customize a DRL training framework that can be used to train the parameters of the GNN.
4.   We carry out a simulation using the OMNet++ platform to validate the proposed routing scheme.

The remainder of this paper is organized as follows: In Section 2, we introduce the related works, which mainly involve the machine learning technologies used in networking. Section 3 introduces the main background and aim of this paper. Section 4 illustrates the DRL framework used to train the neural networks in this paper. Section 5 provides the details of the implementation of the GNN. In Section 6, the experimental setting and results are given. Section 7 concludes the paper.

## 2. Related Works

Supervised learning comprises machine learning methods which are used to classify the input data. During the training process in supervised learning, a labelled data set is required, in order to tell the algorithm what the correct classification outcome is for the calibration of the algorithm parameters. The most prevalent type of supervised learning is the deep neural network (DNN). Studies [12,13] have also proposed the use of DNNs for traffic classification, based on which the routing of the traffic is adjusted. Mao et al. [14] have proposed a routing mechanism using a deep belief network (DBN). Their scheme was designed for the core networks, and edge routers of the network are responsible for the routing path calculation using the DBN. Such supervised learning-based schemes require a large amount of labeled data to carry out the training of the neural networks. However, having a labeled data set is rare in the communication network field, which makes such schemes unpractical.

Reinforcement learning-based schemes represent another type of machine learning algorithm that can be used for routing and traffic engineering. Unlike supervised and unsupervised learning methods, reinforcement learning can directly generate an action for the forwarding behavior and does not require another aided algorithm for routing policies. For example, Hu et al. [15] have used Q-learning to manage the routing based on the energy in the nodes of a wireless sensor network (WSN). Such Q-learning-based methods can only calculate simple input information, as the Q values are stored using value tables. When combined with a DNN, DRL shows more power in dealing with complicated input data. DRL-TE [8] is a DRL-based traffic engineering scheme which was proposed to adjust the proportion of the traffic on multiple candidate paths. Long short-term memory (LSTM) was used in DRL-TE to detect the time-relevance feature of the traffic in the network. Sun et al. [9,10] have proposed the use of DRL to adjust the link weights in a communication network, based on which the global routing of the network is adjusted. Xu [16] has used multi-agent deep deterministic policy gradient (MADDPG) to solve routing problems in a distributed manner. All the DRL-based schemes mentioned above used a new method for missing data estimation or a simple multi-layer perceptron (MLP) to carry out the calculation of the network traffic. As stated in the previous section, such neural networks cannot adapt well to topology changes.

## 3. Routing Optimization

In a communication network, finding an optimal forwarding path for the flows has long been a key consideration. As the configuration of routing policies is an NP-hard problem [17], there have been many studies on the design of practical routing algorithms, most of which are heuristic. The aim of the routing algorithm is to maximize the network performance (e.g., maximizing the throughput or minimizing the end-to-end delay) under a certain network environment.

With the help of SDN, a global view of the network can be obtained from the network. Using the centralized logic of the controller in SDN, the routing algorithms can provide more comprehensive information about the traffic distribution on the network, thus making more accurate decisions on how to steer the network traffic. Figure 1 shows an example of the SDN architecture.

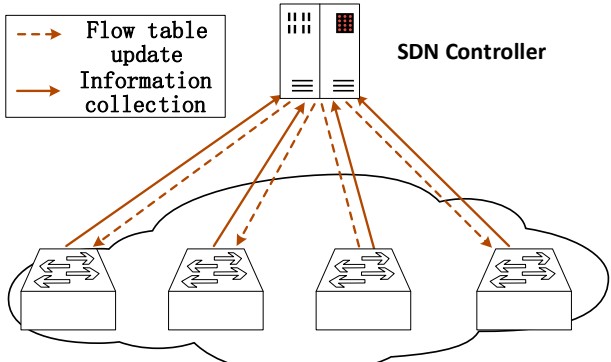

**Figure 1.** SDN architecture.

In the SDN architecture, the controller can adjust and update the flow entries of the SDN-enabled switches (i.e., OpenFlow switches), which determines the forwarding rules of the flows. When the routing paths of a flow are changed in the controller, the path will be denoted with a hop-by-hop forwarding list for this flow. Each hop-by-hop forwarding rule will be converted to the flow entries on the switch, defining which egress port the flow should go through in this switch, and thus changing the routing of this flow.

Routing problems in the network can be regarded in terms of resource allocation from the hardware to the bandwidth requirement of the traffic flows. In a network, there are multiple routing nodes and many links connecting them. Each link has a certain bandwidth capacity and is connected to a port on one routing node. When the bandwidth demand of the traffic is beyond the capacity of the link, congestion will take place, meaning that the data packets are buffered or even dropped in the routing nodes, leading to longer end-to-end delays or packet losses. A good routing algorithm should maximize the capacity of the whole network under certain constraints regarding the end-to-end delay and packet loss rate.

As mentioned above, the routing problem of the communication network is NP-hard. In this paper, we aim to solve this problem through the use of a GNN, using a DRL framework to train the parameters of the GNN. Our proposed automatic routing scheme, which is named AutoGNN, combines GNN and DRL. The overall structure of AutoGNN is shown in Figure 2, which is implemented based on SDN. In AutoGNN, a DRL module is built above the SDN controller. The SDN controller collects network traffic information for the DRL agent, which is reformatted as input information for the DRL algorithm. The output of the DRL agent is used as the action in the SDN controller, and the controller can adjust the routing policies according to this action. The workflow of AutoGNN is shown in Figure 3.

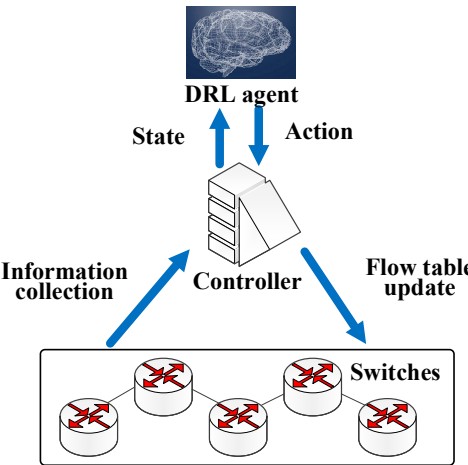

**Figure 2.** The overall structure of AutoGNN.

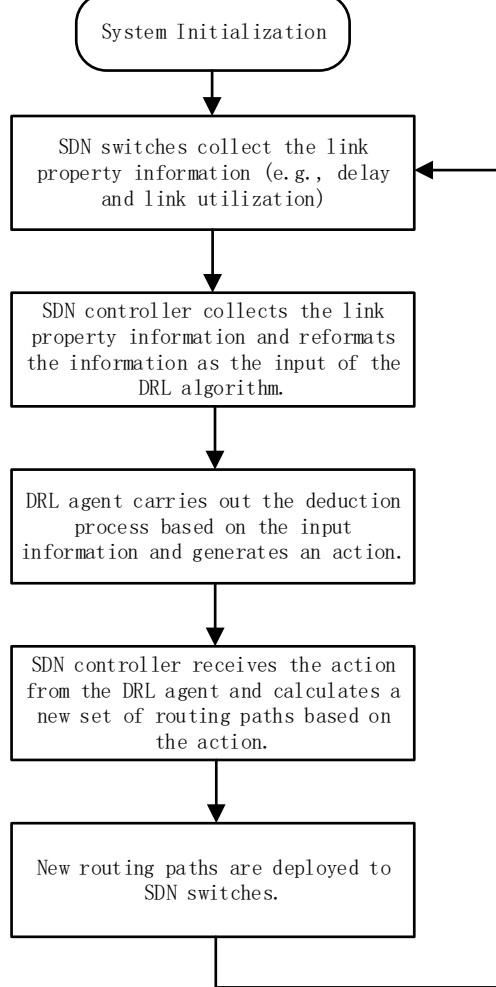

**Figure 3.** The workflow of AutoGNN.

## 4. The DRL Framework of AutoGNN

In this section, we introduce the mechanism of AutoGNN and the corresponding technologies and algorithms used in AutoGNN.

### 4.1. The Basic Framework of RL

DRL is a special group of RL that uses neural networks as function approximators, following the same framework as RL. The basic mechanism of RL is the interaction of the algorithm (also called an agent) with the environment. At each interaction step $k$, the RL agent collects the state $s_k$ of the environment, then a calculation process is carried out within the agent. The agent then generates an output, named an action $a_k$. The action $a_k$ is used to change the behavior of the environment, which leads to a change in the state of the environment and results in the state $s_{k+1}$ in the next interaction step. During this process, the change of the environment is evaluated with a specific standard, and the evaluation of step $k$ is quantized with a value, named the reward $r_k$. The reward value $r_k$ is used to improve the parameters of the RL agent towards obtaining a better performance. The state transition involves some randomness, so the interaction process is usually modeled as a Markov process.

In the learning process of RL, the agent generates the action based on the observed current state of the environment. At first, the parameters of the RL agent are randomly set, thus making it unaware of the transition probability of the environment from $s_k$ to $s_{k+1}$. Then, through interaction with the environment, the parameters of the RL agent evolve. With enough training, the parameter setting of the RL agent will reflect enough information concerning the characteristics of the environment, thus generating near-optimal actions.

The training process of RL is usually measured in episodes. One episode usually contains a series of interaction steps $(s_k, a_k, r_k)$. The training target of the RL agent is to maximize the average accumulated reward in an episode, which is denoted as $E(\sum_{k=0}^{T} r_k)$, where $T$ is the number of steps contained in an episode. To ensure better convergence of the algorithm, the accumulated reward is replaced by $E(\sum_{k=0}^{T} \gamma^k r_k)$ in practice, where $\gamma$ is the discount factor.

### 4.2. The Action Generation Policy

The action of an agent is generated based on a certain function, which is also called a policy. The policy is denoted as $\pi(s, a)$, which denotes the probability of choosing an action $a$ under state $s$. In primitive RL, the policy is usually stored in a lookup table. However, this scenario cannot deal with complicated problems which arise when the number of $(s, a)$ combinations is extremely large. Therefore, the usage of function approximators [18,19] has been proposed. DRL uses a DNN as the function approximator for the agent [14,15], where the parameter set of the neural network is denoted by $\theta$. There are many types of DNN that can be used, such as CNNs [20] and RNNs [8,9]. In this context, the policy of a DRL agent can be written as $\pi_\theta(s, a)$. With the use of a DNN, the DRL agent can process the information from a huge input space, thus endowing DRL techniques with great power to deal with complicated problems.

### 4.3. Policy Gradient Methods

As DRL uses a DNN as the function approximator, the parameters of the neural network need to be trained. In AutoGNN, we use the policy gradient method to train the parameters of the neural network. In policy gradient, the gradient is calculated using Equation (1):

$$\nabla_\theta E_\pi[\sum_{k=0}^{T} r_k] = E_\pi[\sum_{k=0}^{T} \nabla_\theta \log_\pi(s_k, a_k) Q(s_k, a_k)], \tag{1}$$

where $Q(s_k, a_k)$ is an estimate of the quality of the policy. In AutoGNN, we use Equation (2) to calculate the exact quality value of each step.

$$Q_t = \sum_{t'=t_0}^{T} r_{t'} - B_t \tag{2}$$

The main idea of the policy gradient is to collect a trajectory of the interaction between the DRL agent and the environment. Then, the reward values in this trajectory are used

to calculate the gradient [21]. This idea also follows the Monte Carlo method [22], which means that the sample data of the interaction trajectories are used for the calculation of the discounted reward $v_k$ and the estimation of the $Q(s_k, a_k)$ value. After the gradient value is obtained, the parameters of the neural network are updated using Equation (3):

$$\theta \leftarrow \theta + \alpha \sum_{k=0}^{T} \nabla_\theta \log_\pi (s_k, a_k) v_k, \tag{3}$$

where $\alpha$ is the learning rate, which controls the update speed of the parameters. Equation (3) is also the key idea of the REINFORCE method [23], which is an important RL framework. In REINFORCE, the gradient $\nabla_\theta \log_\pi (s_k, a_k) v_k$ points out the direction in which the parameters can be changed towards obtaining a better performance. Putting the value $v_k$ after $\nabla_\theta \log_\pi (s_k, a_k) v_k$ improves the magnitude of the gradient for the performance of the agent, as this means a larger reward has a larger weight in the evolution of the neural network. This is also the reason why the method is named REINFORCE: the action is reinforced with better performance.

The main process of the policy gradient method used in AutoGNN is shown in Algorithm 1. Line 1 randomly initializes the parameters of the neural network. Line 2 defines the total number of iterations for the training process. Lines 3–4 run a sequence of episodes of interactions for the DRL agent. Line 5 calculates the accumulated reward data of each episode. Line 6 resets the update value of the neural networks. Lines 7–8 calculate the baseline value $g_t$, with $g_t = \frac{1}{N} \sum_{i=1}^{N} R_t^i$, to reduce the bias of the accumulated reward value, and lines 9–10 calculate the update value of the neural network parameters. Line 13 regulates the episode length. Finally, Line 14 updates the neural network.

---

**Algorithm 1:** The policy gradient in AutoGNN.

---

The policy gradient in AutoGNN
**Input:** network state and reward;
**Output:** action value;
1: Initialize the neural network parameters;
2 : **For** *iteration* $= 1$ to $M$:
3 : set $l = l_{init}$;
4 : get $(s_1^i, a_1^i, r_1^i, \ldots, s_l^i, a_l^i, r_l^i) \sim \mu_\theta$;
5 : calculate $R_t^i = \sum_{t'=t}^{l} \gamma^{t'} r_{t'}^i$;
6: reset the increment of neural network weights;
7 : **For** $t = 1$ to $l$:
8 : calculate base value $g_t$;
9 : **For** $i = 1$ to $N$:
10 : $\Delta\theta = \Delta\theta + \nabla_\theta \log \mu_\theta(s_t^i, a_t^i)(R_t^i - g_t)$;
11: **End for**
12: **End for**
13 : $l = l + \epsilon$;
14 : $\theta = \theta + \alpha\Delta\theta$;
15: **End for**

---

## 5. GNN Design of AutoGNN

In Section 4, we introduced the DRL framework used in AutoGNN. In this section, we illustrate the type of deep neural network (i.e., the "D" in DRL) used in the DRL framework. Specifically, we chose a GNN as the type of neural network implementation.

### 5.1. Basic Introduction of the Used GNN

AutoGNN uses a GNN to carry out the information calculation for the network topology. GNN is a recently developed type of neural network [24–26], which is better at operating on a graph-structured input. The structure of a GNN is different from that of other neural networks: it includes the ideas of entities (nodes) and links (connections among nodes) above the basic neurons. The input of graph-structured data exhibits a

natural data structure, depending on the nodes and links, and the structure of a GNN is more suitable for processing such data. The initial input graph data are iteratively processed in the GNN, both at the node level and at the level of connections among nodes. Through such iterations, the GNN gradually infers the information contained in the input graph and generates an output that can reflect certain characteristics of this graph.

There are actually various types of GNNs, such as graph convolutional networks (GCNs) [27] and message-passing neural networks (MPNNs) [28]. Different types of GNNs have different specialties. For example, GCNs are good at finding the relationships among nodes, while MPNNs are good at processing the messages transmitted among different nodes. Considering the routing scenario, we employed an MPNN as the main GNN type to process the traffic-related information in AutoGNN.

MPNN performs its function by iteratively carrying out a message-passing operation among the nodes of a graph. Figure 4 shows an example of the message-passing process of MPNN.

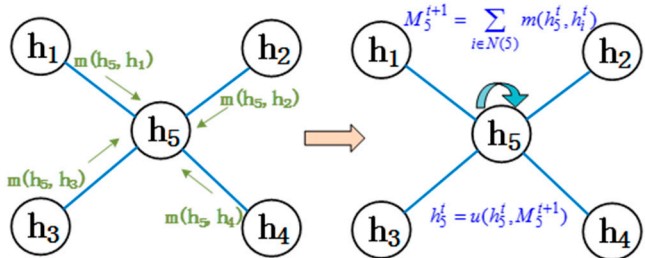

**Figure 4.** An example of the message-passing process.

In the example shown in Figure 4, there are five nodes, and the example shows how node 5 receives and processes the information of its neighbors. First, for the hidden state information of each node, there is a message function $m(\cdot)$ that can calculate one neighbor's information and its local information. Then, all of the outputs of the message functions $m(\cdot)$ are aggregated. The aggregation function is shown in Equation (4):

$$M_5^{t+1} = \sum_{i \in N(5)} m(h_5^t, h_i^t) \tag{4}$$

where $M$ is the aggregated sum of the messages. Then, the hidden state information of each node is updated, according to the aggregated information, using an update function $u(\cdot)$, as in Equation (5):

$$h_5^t = u(h_5^t, M_5^{t+1}) \tag{5}$$

In MPNN, both $m(\cdot)$ and $u(\cdot)$ are implemented using neural networks. The calculation of $m(\cdot)$ and $u(\cdot)$ is executed iteratively. In AutoGNN, the parameters of these neural networks are trained through the policy gradient method of DRL. With this mechanism, the MPNN can process the traffic information of the network topology after enough training.

*5.2. Interface Design of AutoGNN*

In this section, we mainly introduce the interfaces and format of the input/output information for the MPNN used in AutoGNN. As stated before, AutoGNN works through the interaction between the agent and the network environment. At each step, the agent collects network information as input to the MPNN network. In practice, the network information includes the traffic distribution and the link capacity on each link, which is collected by the SDN controller. As for the construction of the MPNN, we implement one node entity of the MPNN corresponding to each of the links of the network. In this way, the MPNN is designed to find the relationships among the links of the network.

After the message-passing process of the MPNN on the input information, the information of each link is summarized into a value as the output. This value corresponds to the

link weight in the network topology. All the link weight values compose the output action of the DRL agent. With the link weight values, a weighted shortest path algorithm is run in the controller to calculate the shortest path for each input flow, thus achieving automatic routing adjustment. When the action of the DRL agent is executed in the network, the routing of the flows is changed, leading to a redistribution of traffic in the network. The redistribution of the traffic then results in a change in the performance of the network (e.g., the average end-to-end delay of the network). The change of the network performance reflects the quality of the policy, which is used to calculate the reward for the current action of AutoGNN. The reward value can be positive, negative, or zero, which reflects whether the action leads to an improvement, degradation, or no influence on the network performance, respectively.

The state information of AutoGNN consists of the remaining link capacity and the target link property. The target link property depends on the target performance evaluation index of the network. For example, if the aim of the routing policy is to minimize the average transmission delay of the whole network, then the target link property is the delay on each link.

Figure 5 shows an example of the format of the state information in AutoGNN. The input features of each link have two values, denoted as Link Feature 1 and Link Feature 2. In practice, we use Link Feature 1 as the placeholder for the remaining link capacity and Link Feature 2 as the placeholder for the target link property (e.g., transmission delay).

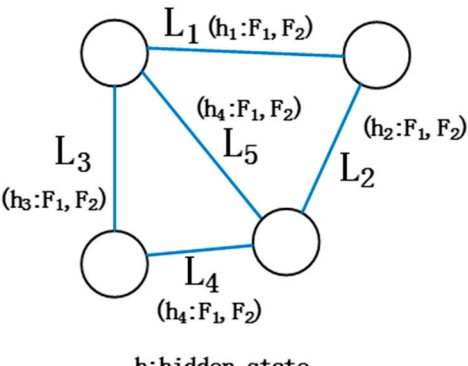

h:hidden state
$F_1$:Link Feature 1
$F_2$:Link Feature 2

**Figure 5.** State processing of AutoGNN.

For simplicity, we describe the main interfaces of AutoGNN in the following:

State: The state information of AutoGNN refers to a list of link features in the network. Each link feature contains the link capacity and the target link property. The link property corresponds to the control target of the network, which can vary according to the target. For example, if the control target is to reduce the overall transmission delay, then the link property is the link delay.

Action: The action value of AutoGNN is a list of link weight values. Each value corresponds to a link weight in the real network topology. The routing path is calculated using a weighted shortest path algorithm based on the link weights. With the dynamic actions generated by AutoGNN, the routing of the network is also changed dynamically.

Reward: The reward of the network is also calculated according to the control target of the network. For example, if the control target is to reduce the overall transmission delay, then the reward can be set based on the variance of the average end-to-end delay of the network.

### 5.3. Implementation of MPNN

In this section, we mainly introduce the implementation of the MPNN in AutoGNN, including the structure of the neural networks for $m(\cdot)$ and $u(\cdot)$ illustrated in Section 5.1.

First, for intuitive understanding of the structure of the MPNN in AutoGNN, we provide an example of the processing of information in Figure 6. When the input states arrive at the input layer of the MPNN, the message-passing procedure iterates on each link's hidden information. Such operations then combine the processed link hidden information together and use a fully connected layer to carry out the combination, which produces the data called messages. The messages of the same entity (corresponding to the links of the network topology) and its neighbors are added. The added information is sent to another neural network, which is implemented with a gated recurrent unit (GRU). The GRU network iteratively calculates the hidden states of the links $T$ times, a process whereby the entities of the MPNN communicate with each other about their "relationships". Finally, after enough calculation, the GRU produces a final hidden value for each entity (i.e., the link of the network topology). The final value is calculated by an NN to a summarized value, which is then used as the output action of AutoGNN.

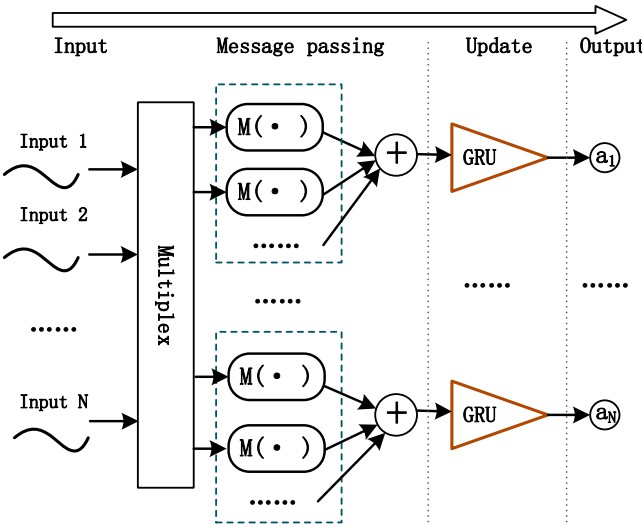

**Figure 6.** The calculation process of AutoGNN.

The calculation process described above is shown in Algorithm 2. This algorithm shows the whole calculation process of one interaction between the DRL agent and the network environment. Line 1 defines the number of iterations for each piece of link information, while Line 2 makes sure that, during each iteration, all links share similar calculation processes. Line 3 is the operation of $m(\cdot)$, which is implemented using a fully connected neural network. Line 4 is the operation of $u(\cdot)$, which is implemented using a GRU. After the update operation of $T$ steps, the final output of each GRU is sent to a simple fully connected neural network to generate a summarized value, which composes the output action of AutoGNN.

---

**Algorithm 2:** The Process of MPNN.

---

The Process of MPNN
**Input:** Link Feature $[h_1, h_2, \dots, h_N]$
**Output:** action value $[a_1, a_2, \dots, a_N]$
1 : For $t = 1$ to $T$:
2 : For $l = 1$ to $N$:
3 : $M_l^{t+1} = \sum_{i \in N(l)} m(h_l^t, h_i^t)$
4 : $h_l^{t+1} = u(h_l^t, M_l^{t+1})$
5: End for
6: End for
7 : For $l = 1$ to $N$:
8 : $a_l \leftarrow FNN(h_l)$
9: End for

---

## 6. Experiment and Evaluation

In this section, we describe an experiment that we carried out in order to compare the performance of AutoGNN with several baseline methods.

### 6.1. Experimental Setup

We implemented the experimental topology based on OMNet++ 4.6. In our simulation, the interaction between the OMNet++ platform and the DRL agent was performed using log files, where the network traffic and routing latencies of all links were all written in log files. The neural networks, together with the training DRL framework, were implemented with Tensorflow 1.12 based on the Python 3.6 software. For the interface between the algorithm and the network environment, we added a self-defined environment following the OpenAI Gym framework [29]. The self-defined environment interacted with the algorithm as a typical Gym environment does, and with the OMNet++ topology through text files. The network topology was implemented based on the 14-node NSFNet topology [30], the 34-node OS3E topology [31], and the 66-node Globenet from the Topology Zoo. For the convenience of comparison among the different schemes, we set all links in the topologies to have the same parameters (i.e., 1 Mbps in bandwidth and no packet loss). All simulations were run on a PC with a CPU of 10700K, 32 GB DDR4 RAM, and 2080Ti GPU.

The AutoGNN agent was trained with 60,000 iterations (i.e., $M$ in Algorithm 1), before being put into use in the routing adjustment. The iteration number $T$ in Algorithm 2 was set to 8, and the optimizer used to update the parameters of the neural networks was the stochastic gradient descent [32] method with Nesterov momentum [33]. The learning rate was set to $5 \times 10^{-5}$.

The test traffic was of three different types, with different average bandwidth requirements. In the test network, each host was a traffic originator, which generated flows to other hosts. The traffic pattern for these traffic originators was composed of two types of flows, namely background periodical flow (PF) and short burst flow (RF) [9,34]. The average throughput of such flows was set according to the three different traffic types. Then, the network environment started, and the hosts continuously generated flows to the destination nodes following a certain traffic pattern. At each pre-set time interval, the AutoGNN agent collected the traffic distribution on the links of the network, which was used as the input state of the GNN. Then, the action generated by the agent was written into a certain text file that was to be read by the OMNet++ simulator. Once the action was read by the simulator, the routing paths of the network were recalculated, leading to a redistribution of the traffic and, thus, a change in the performance metric of the network. The redistribution of the traffic and change of the performance were also written in text files, which were used by the DRL agent for the next state and the reward value.

### 6.2. Baselines

In this section, we mainly introduce the baselines that were used for comparison with AutoGNN. Specifically, we considered the following methods.

Equal-cost multi-path (ECMP): ECMP calculates several candidate paths for each end-to-end communication pair. The paths have roughly the same cost for a flow. Each flow is split evenly into each candidate path.

Scale-DRL: Scale-DRL [35] uses DDPG as the DRL framework to generate automatic routing policies. Similar to AutoGNN, Scale-DRL also controls the link weights for all links in the network and uses a weighted shortest path algorithm to calculate the routing paths.

DRL-TE: DRL-TE [8] is a multi-path scheme for routing. It selects three candidate paths for each end-to-end communication pair. Then, DRL-TE also adopts DDPG as the main DRL framework, with which it generates a set of splitting ratios to decide the proportion of each flow on the candidate paths.

### 6.3. Evaluation and Analysis

Training curve: First, we show the training curve of AutoGNN. As mentioned above, in AutoGNN, we trained the agent with 60,000 iterations. The improvement process of the agent is shown in Figure 7. As shown in Figure 7, as the number of iterations increased, the accumulated reward in each episode also increased significantly. Furthermore, as with most existing DRL schemes, the agent learned more quickly in the beginning, and the trend of increase in the performance tended to slow down as the training process continued. We can see that after about episode 48,000, the reward remained relatively stable.

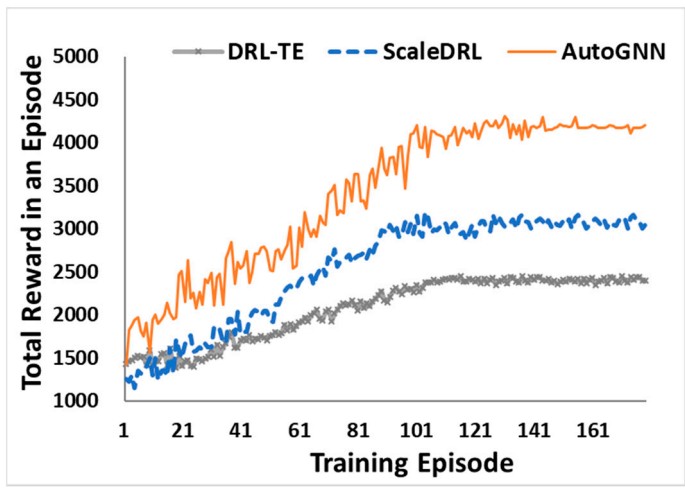

**Figure 7.** Training curve of AutoGNN.

Average end-to-end delay: The average end-to-end delay was chosen as the metric of the performance for the network in this experiment. Therefore, the link feature property of the GNN was set to the value of the link transmission delay (including the delay of the two ports that the link is directly connected to). As we used three sets of traffic with different average traffic demands, and as each flow consisted of periodical flow (PF) and short burst flow (RF), the average end-to-end delay was evaluated with respect to different test scenarios. Figures 8–10 show three experimental results based on different average traffic demands. The traffic demand was calculated based on the average flow throughput versus the link capacity of the originating host of the flow. Therefore, we used 20%, 30%, and 40% to denote the three different traffic demands. In each figure, the x-axis shows the flow component, which is the ratio of the periodical flow in each flow (measured with the average flow size in a period). We can see from Figures 8–10 that as the average traffic demand grew, the end-to-end delay of the network also increased. This was mainly as a larger traffic demand results in a longer queueing delay in the ports of the routing nodes. Specifically, in each figure, we can also see that as the ratio of the periodical flow dropped in the traffic, the machine learning-based solutions had an increasing average end-to-end delay, while the performance of ECMP remained relatively stable. This was mainly because the DRL-based solutions all employed a certain type of GNN (i.e., LSTM in DRL-TE and GRU in Scale-DRL and AutoGNN) to capture the time-relevance of the traffic flows. When there are more unpredictable short burst flows in the traffic, there will be less time-relevance in each flow.

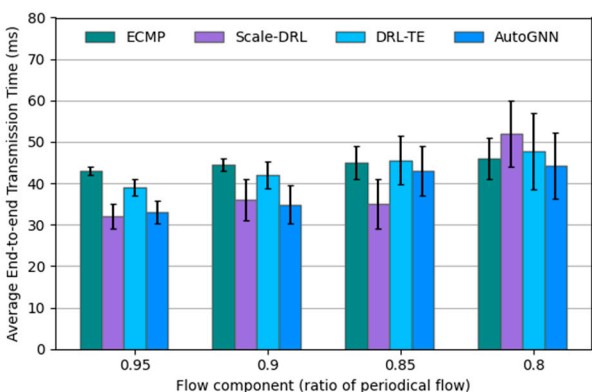

**Figure 8.** The end-to-end delay under 20% traffic demand.

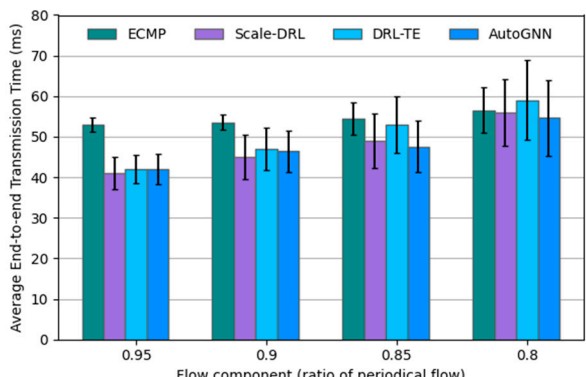

**Figure 9.** The end-to-end delay under 30% traffic demand.

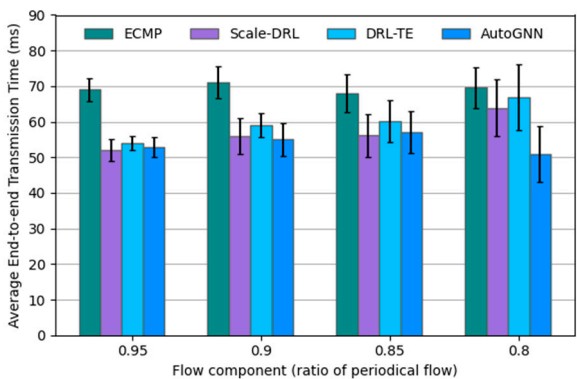

**Figure 10.** The end-to-end delay under 40% traffic demand.

Therefore, such DRL-based solutions have difficulty in accurately modelling the traffic flows, leading to a degradation of the performance when there are more short burst flows. In each bar of Figures 8–10, there is also a line crossing this bar, which denotes the variance of the value. Generally speaking, the DRL-based solutions all had higher variance than ECMP.

Next, we tested the end-to-end delay on the topologies OS3E and Globenet, which had 34 routing nodes and 66 routing nodes, respectively. The results are shown in Figures 11 and 12, respectively. In these tests, the traffic demand was set to 30% of the total capacity of the network. As a result, the larger topology led to a larger end-to-end delay. For these two topologies, AutoGNN also had the best performance. Furthermore, we can see that, for both the OS3E and Globenet topologies, Scale-DRL presented the worst routing performance. This was mainly because, as the topology grows larger, the action space of the DRL algorithm in Scale-DRL suffers from the curse of dimensionality problem, such that Scale-DRL could not properly adjust the routing of the network in these scenarios. DRL-TE

also suffered from the curse of dimensionality problem in these scenarios, but the candidate routing paths were preset in this scheme, such that the traffic could still be contained in relatively good candidate paths, leading to a better performance than Scale-DRL.

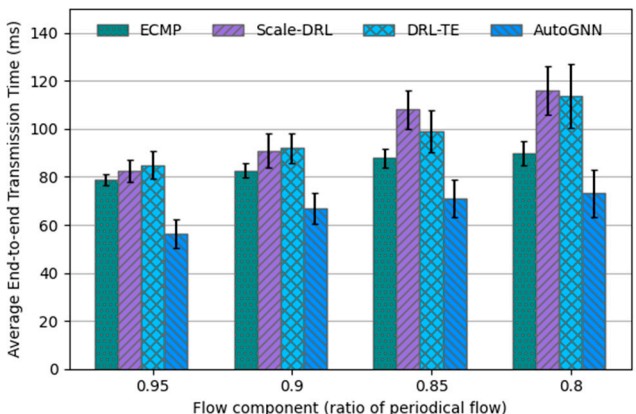

**Figure 11.** The end-to-end delay in OS3E.

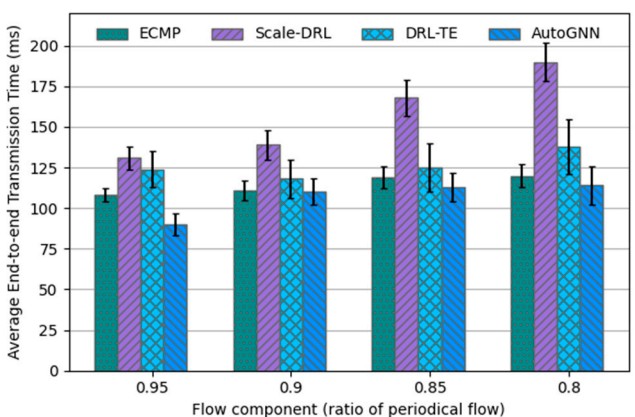

**Figure 12.** The end-to-end delay in Globenet.

In conclusion, considering all of the test scenarios above, we can see that AutoGNN generally presented the best performance among the considered methods in terms of the average end-to-end delay.

Generalization ability: We also evaluated the generalization ability of AutoGNN. As mentioned before, prevalent DRL schemes use a CNN or RNN as the main functional neural network. However, once the training process is completed, such neural networks can only operate on input data with the same dimensions. However, for the network scenario, the shape of the input data is closely related to the network topology, which is likely to change. On the other hand, the network manager would also like to transfer a trained DRL algorithm for other networks, in order to save the training time, which also requires a generalization ability of the DRL algorithm. Therefore, in this test, we evaluated AutoGNN on a series of real-world network topologies, selected from the Topology Zoo data set [36]. Based on the Topology Zoo, we chose topologies whose routing node number was within the range of 10–50 and special topologies that included start or ring topologies were removed. The final number of topologies in our test group was 20.

In this test scenario, we used the parameters of the NSFNet topology as the default parameters for different schemes, such that the neural networks were not retrained for different topologies. For DRL-TE and Scale-DRL, if the new topology had fewer routing nodes, we simply used zero-padding in the input layer to keep the shape of the input data compatible for the neural networks; if the new topology had more routing nodes, we just

added the number of input/output neurons in the network and set the weights of the newly added neurons to random values.

We ran 1000 steps for each topology and collected the average end-to-end delay. The results are shown in Figure 13, where the x-axis shows the ordinal number of different topologies (i.e., orders according to the deviation of the node number compared to NSFNet) and the y-axis shows the average delay, compared to the delay values of corresponding schemes in the NSFNet (positive value means an improvement of the performance and negative value means a degradation of the performance). As shown in Figure 13, AutoGNN and ECMP maintained relatively good performance under different topologies, where AutoGNN performed better than ECMP. However, both DRL-TE and Scale-DRL suffered from a significant degradation of performance under different topologies. Furthermore, the higher the deviation of the new topology compared to NSFNet, the more serious the degradation. This demonstrates the generalization ability of AutoGNN, which mainly relies on the generalization ability of the GNN network.

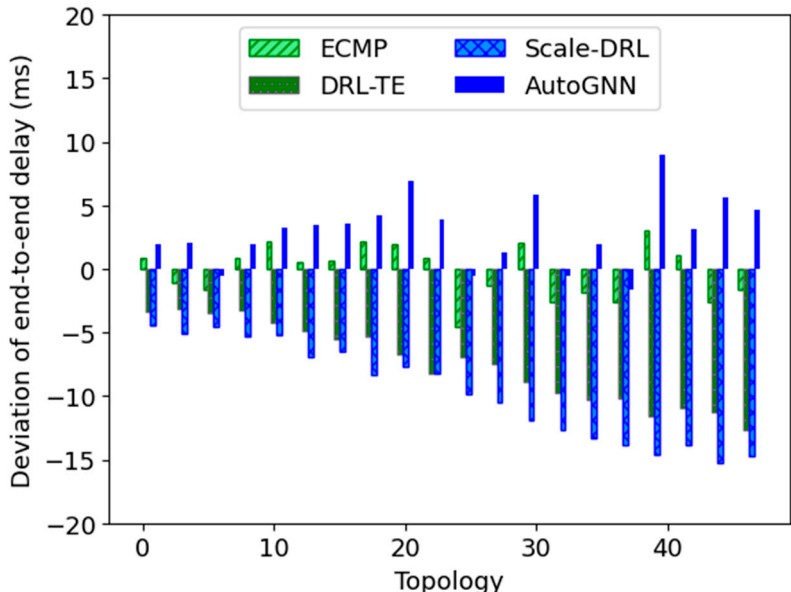

**Figure 13.** Generalization ability (positive values denote an increase in end-to-end delay and negative value denotes the decrease in end-to-end delay).

Application: In this test, we present an application scenario in which the advantages of AutoGNN can be exploited. This scenario was conducted using a Geant2 topology and shows how AutoGNN performs against link failures. The link failures take place in the original topology, thus changing the traffic distribution of the network. When there is a link failure in the network topology, the routing policy must find a backup routing path for all the flows that were transmitted through this link. We carried out this test by randomly cutting off the links of the original network topology, where the number of cut links ranged from 0 to 8. The AutoGNN agent was trained in the original AutoGNN topology and, in each link cutting test, the AutoGNN agent was run for 1000 steps to ascertain the performance of the network. The traffic in the test was the same as that used in Figure 6, with the ratio of periodical flow set to 0.9. The results are shown in Figure 14.

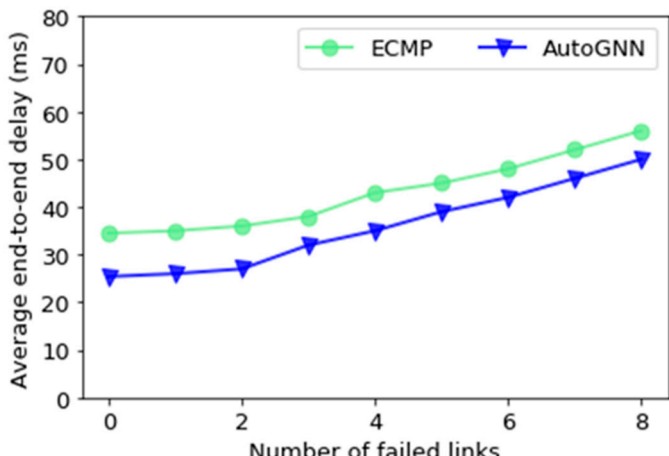

**Figure 14.** Test of AutoGNN under link failure.

As Figure 14 shows, when the number of failed links increased, the average end-to-end delay of the network also increased. This was mainly because fewer links will decrease the overall capacity of the network, such that there will be more congestion under the same traffic demand. However, compared to ECMP, AutoGNN always showed an advantage, demonstrating the better generalization ability of AutoGNN. This application scenario validated the robustness of AutoGNN against network failures.

Efficiency: Routing latency is a significant parameter, so we expect the routing policy to be generated as quickly as possible. As different routing schemes may use different hardware to perform the policy calculation (e.g., neural network-based solutions prefer to run on GPU), we compared the policy generation time of different schemes to assess the efficiency of different schemes. Figure 15 shows the run-time of different schemes under the OS3E topology in our experimental platform. As shown in Figure 15, ECMP had the lowest run-time, as it is a simple routing policy. ScaleDRL, DRL-TE, and AutoGNN all required a little more time than ECMP, as they carry out calculation using neural networks (which are mainly performed in GPU). Considering the improvement in routing performance, we believe that the small extra time cost of AutoGNN is acceptable.

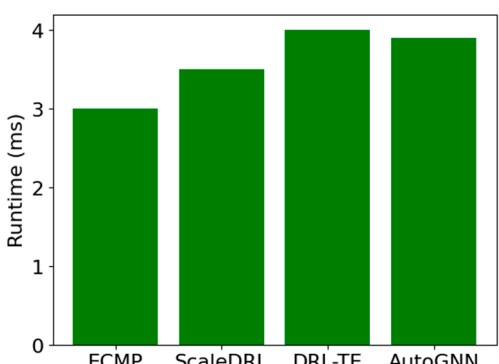

**Figure 15.** Test of AutoGNN run-time under link failure.

## 7. Conclusions

In this paper, we proposed an automatic routing scheme, AutoGNN, based on the combination of a GNN and a DRL. The usage of a GNN endows AutoGNN with two advantages. First, the GNN can better process the graph-like traffic information of a communication network compared with other neural networks, such as CNNs or RNNs. Second, the GNN has good generalization ability, such that a trained AutoGNN agent can also perform well under various topologies, and shows more robustness against link failures compared with other DRL-based routing schemes. We also demonstrated the

effectiveness of using a GNN in solving network problems, which may pave the way for other studies that attempt to solve networking problems through the use of machine learning technologies.

**Author Contributions:** B.C.: Drafting the work and literature search; D.Z.: Methodology; Y.W.: Interpretation of data; P.Z.: Scheme design and data collection. All authors have read and agreed to the published version of the manuscript.

**Funding:** Ministry of Science and Technology: 2020YFB1804800.

**Conflicts of Interest:** The authors declare no conflict of interest.

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
