# Peer review of "An Approach to Combine the Power of Deep Reinforcement Learning with a Graph Neural Network for Routing Optimization"

_electronics, doi:10.3390/electronics11030368_

Round 1

Reviewer 1 Report

the paper present graph based technique to address the routing optimization in Networking. The paper idea is novel and literature written covers all the important papers in the field. however, few points must be highlighted

1-The author must add the AutoGNN network architecture and network flow diagram to help user better understand the concept.

2- Time complexity of proposed work must be compared with state of art as in routing latency is major concern.

3- in results, no simulation figure is presented only results are explained. Add few simulation figures of OMNET++. 

4- the authors discussed the integration of OMNET++ with CNN network. It should be explained in little detail.

5-the SDN, RNN are mentioned in the paper but not cited. Add citation of previous work discreetly.

Reviewer 2 Report

This paper studied the technology of Combining the Power of Deep Reinforcement Leaning with Graph Neural Network for Routing Optimization. It is a little meaningful. The quality is just so-so. Major revision should be done for this version of the paper as follows:
1) More achievements on this topic of Combining the Power of Deep Reinforcement Leaning with Graph Neural Network for Routing Optimization should be added for section 1 and 2. Some out-of-date idea from out-of-date references in Section "introductions"  and "related work" should be revised.
2) Mathematics modeling to supprot and analyze the method is not enough. The method of Combining the Power of Deep Reinforcement Leaning with Graph Neural Network for Routing Optimization should be detailedly descirbed step by step. The cost analysis of the method of Combining the Power of Deep Reinforcement Leaning with Graph Neural Network for Routing Optimization should be added. 
3) The figures form Figure 7 to 12 must be re-done with different legend to identify the line not only by the color. More compelx comparison curve figures to do detailed discussion with the relative methods should be supplied in the Section.
4) The format of the reference should be improved. Some references are out-of-date, so these references before 2011 should be deleted. At the same time, many important recent references are missing, which can support the idea of this paper,  the following references should be totally added in the Section "References":   

[1]Xiaohuan Liu. Novel Best Path Selection Approach Based on Hybrid Improved A* Algorithm and Reinforcement Learning. Applied Intelligence,2021,51(9):1–15.   DOI:10.1007/s10489-021-02303-8

[2]Xiaohuan Liu. A path planning method based on the particle swarm optimization trained fuzzy neural network algorithm.Cluster Computing,2021,24(1):1-15.   DOI:10.1007/s10586-021-03235-1

[3]Chang-le Gong,. A new algorithm of clustering AODV based on edge computing strategy in IOV. Wireless Networks, 2021, 27(4):2891-2908.  DOI:10.1007/s11276-021-02624-z

[4]ZHANG T. A New Method of Data Missing Estimation with FNN-Based Tensor Heterogeneous Ensemble Learning for Internet of Vehicle[J]. Neurocomputing, 2021,420(1):98-110.   DOI:10.1016/j.neucom.2020.09.042.

[5]WANG X, SONG X D. A Novel Approach to Mapped Correlation of ID for RFID Anti-collision[J]. IEEE Transactions on Services Computing, 2014,7(4):741-748.

[6]YANG J N,MAO G Q. Optimal Base Station Antenna Downtilt in Downlink Cellular Networks[J].IEEE Transactions on Wireless Communications, 2019,18(3):1779-1791.   DOI:10.1109/TWC.2019.2897296

[7]Cui Y Y, Zhang T. New Quantum-Genetic Based OLSR Protocol (QG-OLSR) for Mobile Ad hoc Network. Applied Soft Computing, 2019,80(7):285-296.  DOI:10.1016/j.asoc.2019.03.053

[8]GE H. New Multi-hop Clustering Algorithm for Vehicular Ad Hoc Networks[J]. IEEE Transactions on Intelligent Transportation Systems, 2019, 20(4):1517-1530.     DOI:10.1109/TITS.2018.2853165

[9]CHEN J Q, MAO G Q. Capacity of Cooperative Vehicular Networks with Infrastructure Support:Multi-user Case [J]. IEEE Transactions on Vehicular Technology, 2018,67(2):1546-1560.     DOI:10.1109/TVT.2017.2753772

[10]ZHANG D G, LI G, ZHENG K. An energy-balanced routing method based on forward-aware factor for Wireless Sensor Network[J]. IEEE Transactions on Industrial Informatics, 2014,10(1):766-773.

[11]CUI Y Y. Novel Method of Mobile Edge Computation Offloading Based on Evolutionary Game Strategy for IoT Devices[J]. AEU-International Journal of Electronics and Communications, 2020,118(5):1-13.  DOI:10.1016/j.aeue.2020.153134

[12]CHEN L, ZHANG J. A multi-path routing protocol based on link lifetime and energy consumption prediction for mobile edge computing[J]. IEEE Access,2020,8(1):69058-69071.   DOI:10.1109/ACCESS.2020.2986078

[13]PIAO M J, ZHANG T, New Algorithm of Multi-Strategy Channel Allocation for Edge Computing[J]. AEUE - International Journal of Electronics and Communications,2020,126(11):1-15.       DOI:10.1016/j.aeue.2020.153372

5) Add the sub-section “Discussion”. In this sub-section, in order to support the new idea of this paper,  the relative comparison or discussion should be added on the technology between this paper and the above given references.

6) The writing format of the paper should be revised.
7) Check and polish the whole paper.

Round 2

Reviewer 2 Report

The authors have revised the paper, but there is some shortcoming in this version, so major revision should be continuously done for this version of the paper as follows:

1) The title of the paper should be improved. The keyword "Approach" or "Mechanism" or so on should be added for the title. 
2) The theory to support the Deep Reinforcement Leaning with Graph Neural Network for Routing Optimization is not enough in the relative sectionsm such as the lemma or theorem. Mathematics modeling to support and analyze the method or mechanism is not enough. The cost analysis should be added for the method or the mechanism.
3) More comparison curve figures to do detailed discussion with the relative methods should be supplied in the Section 6.
4) The format of the reference should be improved. At the same time, many important recent references are missing, which can support the idea of this paper,  the following references should be totally added in the Section "References":   

[1]CHEN J Q. A Topological Approach to Secure Message Dissemination in Vehicular Networks[J]. IEEE Transactions on Intelligent Transportation Systems, 2020,21(1):135-148.     DOI:10.1109/TITS.2018.2889746

[2]ZHANG T. A Kind of Novel Method of Power Allocation with Limited Cross-tier Interference for CRN[J]. IEEE Access, 2019,7(1):82571-82583.     DOI:10.1109/ACCESS.2019.2921310

[3]LIU X H. A New Algorithm of the Best Path Selection based on Machine Learning[J]. IEEE Access, 2019,7(1):126913-126928.     DOI:10.1109/ACCESS.2019.2939423

[4]ZHAO P Z, CUI Y Y. A New Method of Mobile Ad Hoc Network Routing Based on Greed Forwarding Improvement Strategy. IEEE Access, 2019,7(1):158514-158524.     DOI: 10.1109/ACCESS.2019.2950266

[5] LIU S. Novel Unequal Clustering Routing Protocol Considering Energy Balancing Based on Network Partition & Distance for Mobile Education[J]. Journal of Network and Computer Applications, 2017,88(15):1-9. DOI:10.1016/j.jnca.2017.03.025

[6]ZHOU S. A low duty cycle efficient MAC protocol based on self-adaption and predictive strategy[J]. Mobile Networks & Applications, 2018,23(4):828-839.     DOI: 10.1007/s11036-017-0878-x

[7]WU H, ZHAO P Z. New Approach of Multi-path Reliable Transmission for Marginal Wireless Sensor Network[J]. Wireless Networks,2020,26(2):1503–1517.  DOI: 10.1007/s11276-019-02216-y

[8]LIU S. Adaptive Repair Algorithm for TORA Routing Protocol based on Flood Control Strategy[J]. Computer Communications, 2020,151(1):437-448.   DOI:10.1016/j.comcom.2020.01.024

[9] LIU S. Novel Dynamic Source Routing Protocol (DSR) Based on Genetic Algorithm-Bacterial Foraging Optimization (GA-BFO)[J]. International Journal of Communication Systems,2018,31(18): 1-20      DOI: 10.1002/dac.3824

[10]ZHENG K, ZHAO D X. Novel Quick Start (QS) Method for Optimization of TCP[J]. Wireless Networks, 2016,22(1):211-222.

[11]ZHU Y N. A new constructing approach for a weighted topology of wireless sensor networks based on local-world theory for the Internet of Things (IOT)[J]. Computers & Mathematics with Applications, 2012,64(5):1044-1055

[12]Chen-hao Ni, Jie Zhang. A Kind of Novel Edge Computing Architecture Based on Adaptive Stratified Sampling. Computer Communications,2022,183(2022):121-135.  DOI:10.1016/j.comcom.2021.11.012

5)In the sub-section “Discussion”, in order to support the new approach or idea of this paper,  the relative comparison or discussion should be added on the technology between this paper and the above given references.

6) The writing format of the paper should be improved.
7) Check and polish the whole paper once again.

Round 3

Reviewer 2 Report

The authors have revised the paper, but there is some shortcoming in this verison of the paper, so minor revision should be done for this version of the paper as follows:
1) The sub-section number should be revised, such as section 2. Some more relative introductions should be added in this section. 
2) some lemmas or theorems should be added to supprot and analyze the method on Deep Reinforcement Leaning with Graph Neural Network for Routing Optimization in section 3 or 4.
3) Some curve figures should be added to compare the perfermances on Deep Reinforcement Leaning with Graph Neural Network for Routing Optimization. At least 5 compared ojbects should be shown for each curve figure. 
4) Some references are out-of-date, so these references before 2011 should be deleted. At the same time, some important recent references are missing, which can support the idea of this paper,  the following references should be totally added in the Section "References":   

[1]Ran Su, Yixuan Huang. SRDFM: Siamese Response Deep Factorization Machine to improve anti-cancer drug recommendation. Briefings in Bioinformatics, 2022,2:1-13.  DOI:10.1093/bib/bbab534

[2]WANG J X, Hong-rui Fan. New Method of Traffic Flow Forecasting Based on Quantum Particle Swarm Optimization Strategy for Intelligent Transportation System[J]. International Journal of Communication Systems, 2020,33(10):1-13.    DOI:10.1002/dac.4647

[3]CHEN C, CUI Y Y. New Method of Energy Efficient Subcarrier Allocation Based on Evolutionary Game Theory. Mobile Networks and Applications, 2021,26(2):523-536.       DOI: 10.1007/s11036-018-1123-y

[4]NIU H L. Novel PEECR-based Clustering Routing Approach[J]. Soft Computing, 2017, 21(24): 7313-7323.  DOI: 10.1007/ s00500-016-2270-3

[5] Zheng Ke, Zhang Ting. A Novel Multicast Routing Method with Minimum Transmission for WSN of Cloud Computing Service. Soft Computing, 2015,19(7):1817-1827.

[6]GAO J X. Novel Approach of Distributed & Adaptive Trust Metrics for MANET[J]. Wireless Networks, 2019,25(6):3587–3603.      DOI:10.1007/s11276-019-01955-2

[7]ZHANG T. Novel Self-Adaptive Routing Service Algorithm for Application of VANET[J]. Applied Intelligence, 2019, 49(5):1866-1879.  DOI: 10.1007/s10489-018-1368-y

[8]ZHANG T. Novel Optimized Link State Routing Protocol Based on Quantum Genetic Strategy for Mobile Learning[J]. Journal of Network and Computer Applications, 2018,2018(122):37-49.  DOI:10.1016/j.jnca.2018.07.018

5) In the sub-section “Discussion”, in order to support the new idea of this paper, the relative comparison or discussion should be added on the technology between this paper and the above given references.

6) The writing format of the paper should be based on the journal.
7) Check and polish the whole paper once again.
